# Surface Modification of Iron Oxide-Based Magnetic Nanoparticles for Cerebral Theranostics: Application and Prospection

**DOI:** 10.3390/nano10081441

**Published:** 2020-07-24

**Authors:** Yanyue Wu, Zhiguo Lu, Yan Li, Jun Yang, Xin Zhang

**Affiliations:** 1National Key Laboratory of Biochemical Engineering, Institute of Process Engineering, Chinese Academy of Sciences, Beijing 100190, China; yywu@ipe.ac.cn (Y.W.); zglu18@ipe.ac.cn (Z.L.); liyan310@ipe.ac.cn (Y.L.); 2School of Chemical Engineering, University of Chinese Academy of Sciences, Beijing 100049, China

**Keywords:** theranostics, iron oxide magnetic nanoparticles, blood-brain barrier, surface coating, target therapy

## Abstract

Combining diagnosis with therapy, magnetic iron oxide nanoparticles (INOPs) act as an important vehicle for drug delivery. However, poor biocompatibility of INOPs limits their application. To improve the shortcomings, various surface modifications have been developed, including small molecules coatings, polymers coatings, lipid coatings and lipopolymer coatings. These surface modifications facilitate iron nanoparticles to cross the blood-brain-barrier, which is essential for diagnosis and treatments of brain diseases. Here we focus on the characteristics of different coated INOPs and their application in brain disease, particularly gliomas, Alzheimer’s disease (AD) and Parkinson’s disease (PD). Moreover, we summarize the current progress and expect to provide help for future researches.

## 1. Introduction

Magnetic nanoparticles have drawn worldwide attention for their nanoscale physicochemical properties, especially in theranostics [1,2,3,4,5,6,7,8]. Combining diagnosis with therapy, magnetic nanoparticles have a pivotal role in the field of medicine [9,10,11,12,13]. Iron, as the most abundant organic metallic element, has emerged as the most appealing candidate [14,15]. Magnetite (Fe_3_O_4_) and maghemite (γ-Fe_2_O_3_) in particular, display size-control superparamagnetism, and are used in magnetic resonance imaging (MRI). There have been two generations of iron contrast agents (CAs) for MRI, the first one has diagnostic capability only while the new one, combining diagnosis with therapy, has multiple functions. The new one loaded with therapeutic agents after surface coating to facilitate MRI guided drug delivery, gene delivery, photothermal therapy (PTT), photodynamic therapy (PDT) or magnetic hyperthermia has gained attention [16] (Figure 1). By grafting biorecognition molecules (ligands) onto the surface of nanoparticles, active targeted therapy expands the application of new-generation CAs [17,18].

As a protective barrier for the central nervous system, the blood-brain barrier (BBB) limits uptake of drugs in brain cells [19,20]. Because of the tight cell junction, low efficiency of endocytosis, expression of outwardly drug efflux transporters, and high level of drug-metabolizing enzymes of BBB [21], it is difficult for drugs to pass through BBB. Ligand-based active targeted therapy opens a new way to improve the effect of drug treatment [22]. Many researches have taken up ligand-targeted surface-modified iron nanoparticles (INOPs) in brain diseases. Here the review summarizes the application of INOPs in cerebral theranostics, particularly in glioblastoma multiforme (GBM), Alzheimer’s Disease and Parkinson’s disease.

### 1.1. Characteristics of Iron-Based Nanoparticles

Co-precipitation [23], thermal decomposition [24], hydrothermal reactions [25], microemulsion [26], sol-gel [27] are common methods to produce iron nanoparticles. Of all, thermal decomposition and hydrothermal reactions perform well in uniformity and shape [28]. Iron nanoparticles with controlled size can be synthesized from several to dozens. Iron nanoparticles have stronger magnetization with a larger size in some extent in theory [29]. For example, Y Jun found 4 nm, 6 nm, 9 nm, 12 nm magnetic Fe_3_O_4_ nanosphere’s transverse relaxivities R_2_ value, synthesized through thermal decomposition of iron acetylacetonate (Fe(acac)_3_), changed from 78 to 106, 130, and 218 mM^−1^s^−1^ [30] (Figure 2). Thermal decomposition of iron carbonyl (Fe(CO)_5_) formed 8 nm, 23 nm, 37 nm and 65 nm iron nanoparticles with R_2_ 174, 204, 240, 249 mM^−1^s^−1^ respectively [31]. Different shapes of INOPs can also be synthesized by changing temperature, pH, solvent, concentration of the precursors to improve relaxation rate and meet different drug delivery needs [32,33]. J Mohapatra found the R_2_ values of Fe_3_O_4_ nanorods with lengths of 30, 40, 50, 60 and 70 nm are 312, 381, 427, 545 and 608 mM^−1^s^−1^ respectively [34]. N Lee synthesized 22 nm water-dispersible ferrimagnetic iron oxide nanocubes with R_2_ 761 mM^−1^s^−1^ showing superior T_2_ contrast effect [35]. Moreover, 30 nm octapod iron oxide nanoparticles exhibit R_2_ with 680 mM^−1^s^−1^ [36]. Most synthesized iron nanomaterials, prepared in organic solvents, are hydrophobic. However, they need to be water-soluble for medical application. Surface modification is one of the tools to do this. Moreover, surface materials have great influence on hydrodynamic diameter and surface charge, crucial in cell uptake [37]. They also provide functional groups for drugs loading. Surface coatings endow nanoparticles with specific functions, such as lysosomal escape and various stimuli-responsive release, which will be discussed in detail in the part of “Application of Different Coating of INOPs in Cerebral Theranostics”.

### 1.2. Treatment Difficulties and Current Solutions of Brain Diseases

BBB is tightly composed of neuronal pericytes, perivascular astrocytes, and brain capillary endothelial cells (BCECs), limiting drug absorption (Figure 3B) [11]. High levels of active efflux transport proteins, including p-glycoprotein (P-gp), multidrug resistance protein-1 (MRP-1), and breast cancer resistance protein (BCRP) are expressed (Figure 3A). There are several mechanisms for non-invasive drug delivery: (1) passive transportation across the membrane along the concentration gradient through paracellular or transcellular (Figure 3D); (2) receptor mediated endocytosis (Figure 3E), such as transferrin receptor (TfR) [38], lactoferrin receptor (LfR) [39], lipoprotein receptor-related protein (LRP) [40], membrane γ-glutamyl transpeptidase (GGT) [41]; (3) adsorptive mediated endocytosis (Figure 3F), such as trans-activating transcriptional (TAT) peptides [42]; (4) carrier mediated transport (Figure 3C), such as amino acid transporter 1 (EAAT1) [43] and glucose transporters (GLUT1) [44,45,46]. Small lipophilic molecules, less than 400–500 Da and less than nine hydrogen bonds, can cross BBB by passive transportation. However, most larger nutrients cross BBB with the aid of carrier molecules [20].

## 2. Application of Different Coating of INOPs in Cerebral Theranostics

Brain diseases are hard to treat because of BBB, and many iron nanoparticles are designed to overcome this difficulty [47,48,49]. Of all, gliomas [50,51,52,53], Alzheimer’s disease (AD), Parkinson’s disease (PD) [54,55,56] are widely studied. Glioma, especially glioblastoma multiforme, has poor prognosis after common surgical resection, adjuvant chemotherapy, or novel adjuvant therapy combination. Nanomedicine opens a new way for the treatment of glioma in overcoming BBB. Neurodegenerative diseases, as central nervous system (CNS) diseases, exhibit progressive cognitive impairment, loss of memory [57]. The cause of the disease is complex, mainly reported to be abnormal accumulation of protein in the brain [58]. For AD, FDA-approved treatments focus on cholinesterase (donezepil) [59], and N-Methyl-D-Aspartic Acid (NMDA) receptors(memantine) [60]. Many researches on reducing Aβ plaque-secretase modulators (BACE1 [60,61,62,63]), hyperphosphorylation tau protein [64,65], inflammation [66,67], other oxidative molecules (metal ion [68], peroxide [69]) have been studied. For PD, α-syn abnormal aggregation causing impaired motor function with slow movements and tremor [70], studies mainly concentrate on dopamine [71]. Combining those mechanism molecules with ligands overcoming BBB, iron nanoparticles make a great contribution to cerebral disease treatment and diagnosis.

To change the surface charges and hydrophobicity it is possible to increase biocompatibility, endow INOPs with the ability to target and release on demand (Table 1). Different kinds of surface coatings are developed. Here we will discuss the application of different coated INOPs in the treatment of AD, PD, and glioma.

### 2.1. Small Molecules Coating

The presence of numerous OH^−^ on the surface of INOPs provides a chance for the attachment of small molecules and surfactants, which can maintain a small hydrodynamic radius, retain the original magnetic properties and improve hydrophilicity at the same time [72]. The ligand- and phase-exchange reactions are the most conventional methods for small molecules modification. Catechol, carboxylic, phosphate, sulfate, and citrate [73] have strong binding force with INOPs so that they can exchange with the surface organic groups of INOPs like oleic acid or oleylamine. They also perform well as tail of polymers like poly (ethylene glycol) (PEG), polyethyleneimine (PEI).

#### 2.1.1. Silicon

Silicon is one of the most widely studied small molecule coatings. Silane can be covalently bound to the surface of INOPs using the reaction of alkoxysilane functions (–Si–O–R, where R is commonly –CH_3_ or –CH_2_–CH_3_) with hydroxyl group of INOPs [74,75]. Further crosslinking events produce a thin inorganic silica layer around the particles. There are four approaches to prepare INOPs@SiO_2_, Stöber method, microemulsion, aerosol pyrolysis, and methods based on sodium silicate solution [28]. Silica coating turns hydrophobic iron nanoparticles into hydrophilic ones [76], reducing aggregation and improving stability. And surface silanol groups facilitate grafting of functional groups [77]. Modified silane, such as 3-aminopropyltriethyloxysilane (APTES), p-aminophenyl trimethoxysilane mercaptopropyltriethoxysilane (MPTES), and 2-(carboxymethylthio) ethyltrimethylsilane, is often used for transferring –NH_2_, –SH and –COOH groups to naked iron oxide NPs, respectively [78], which is suitable for further modification with drugs or targets. APTES was beneficial to maintain the morphology of Fe_3_O_4_ while MPTES modification resulted in a slight decrease in saturation magnetization [79]. Silane can also perform well as the tail of polymers instead of forming a silicon coating.

The result of thermotherapy using dextran and amino-silane coated INOPs revealed that amino-silane coated INOPs prolonged the survival time by 4.5-fold over dextran-INOPs in malignant glioma rat [80]. PEG with terminal silane coated INOPs were connected to Alexa Fluor 680 (AF680) fluorochrome using chlorotoxin (CTX) as target molecule, showing no difference in cell toxicity compared with untreated group in C6 glioma cell [81]. PEG-INOPs-AF680/CTX reduced the expression of matrix metalloproteinase 2 (MMP-2) by 40% in glioma cell surface, which is an essential component in the glioma cell invasion pathway. Further crosslinking of silane produced SiO_2_ coated INOPs. INOPs@SiO_2_ covalently modified with interleukin-6 receptor targeting peptides (I6P7) was used as MRI probe for glioma. Its positive charge was favorable for BBB crossing, increasing the accumulation in the tumor region [82]. Gold and iron oxide NPs in the hollow of silica golf balls (MGNS) was made to deliver Dox before encapsulation in heat and pH sensitive polymer poly (N-isopropylacrylamide co methacrylic acid) (p (NIPAM-co-MAA)), enhancing drug uptake of neurons in the magnetic field [83].

#### 2.1.2. Catechol

Inspired by natural phenomena, catechol-derived ligands for INOPs have been developed. Dopamine and its modifications are important catechol. Rajh et al. believed dopamine assembled on Fe_2_O_3_ nanoparticles by bidentate chelating interaction [84]. In weak alkaline solution, dopamine is prone to self-oxidation, resulting in attached PDA coating. PDA coating exhibits a special zwitterionicity: it is positively charged at low pH while it turns to be negative at high pH, which has potential application in drug delivery [85]. Moreover, dopamine can also have functions as the tail of other polymers.

Caffeic acid (3,4-dihydroxy-cinnamic acid, CA), with positive biological effects such as anti-oxidant, anti-inflammatory, anti-HIV, anti-tumor, and anti-metastatic effects was used as coating of INOPs due to its strong affinity for INOPs, which could make anti-oxidant drugs keep for at least 45 min in the brain [86]. It plays dual functions as both small surface coating and anti-oxidant drug. Nitrodopamine anchored INOPs were made to deliver anti-cancer drug Dox and a macrocyclic chelating agent (1,4,7-triazacyclononane-N, N′, N″-triacetic acid, NOTA) before conjugation with targeting peptide-cyclic arginine-glycine-aspartic acid (cRGD) peptide (Scheme 1). The transverse (R_2_) relaxivities of nanoparticles (101.9 mM^−1^s^−1^) was slightly lower than that of FDA approved dextran-coated Feridex (111.5 mM^−1^s^−1^). Nanoparticles exhibited a higher cellular uptake, leading to higher cytotoxicity [87].

### 2.2. Polymers Coating

INOPs coating can be obtained through lots of approaches, such as in situ coating, ligand exchange, post-synthesis adsorption, post-synthesis end grafting [88], and so on. To date, many polymers coatings have been developed via surface coverage or forming micelles. Dextran, chitosan, polyethylene glycol (PEG), polyvinyl alcohol (PVA), polydopamine (PDA), polysaccharide, polyethyleneimine (PEI), polyvinylpyrrolidone (PVP), and polyamidoamine (PAMAM) are common surface coverage polymers used in the coating of INOPs [28]. Dextran, PEG, or their modifications coated INOPs have been clinically approved or approved in clinical test (Figure 4a).

To combine the advantages of different polymers, copolymers are developed. For example, copolymers of PEG and PEI have both the ability to load gene and prolong half-life in blood [89]. Copolymers of PLGA and PEG can help nanoparticles escape from the endo-lysosomal compartment to the cytoplasmic compartment and reduce hydrophobicity of PLGA [90]. Amphiphilic block copolymers have facilitated the preparation of iron-based micellar drug carrier systems by self-assembly of the copolymers in aqueous solution to generate polymeric micelles with diameters of 10–100 nm. The hydrophobic blocks are in the core with hydrophilic block forming shell, reducing interactions with proteins and prolonging blood circulation time. Such micelle provides hydrophobic space for hydrophobic iron nanoparticles (Figure 4). If the concentration of copolymer is above the critical micelle concentration (CMC), micelles are formed in various shapes, and the shapes depend on the temperature, polymer concentration, pH, ionic strength of the solution, and so on [4]. Various amphiphilic block copolymers have been developed to coat iron nanoparticles by copolymerization of common polymers, such as PEG, poly(carboxybetaine) (PCB), poly(lactide-co-glycolide (PLGA), poly(ethylene oxide) (PEO), poly(3-caprolactone)(PCL), poly (l-lactic acid) (PLLA), N-(2-hydroxypropyl) methacrylamide (HPMA), poly(e-caprolactone) etc. Moreover, stimuli-sensitive amphiphilic block copolymers have been designed to control release of drugs.

#### 2.2.1. Dextran

Dextran-coated superparamagnetic iron oxide nanoparticles (SPIONs) were described in 1982 by in-situ technology and have been approved by Food and Drug Administration (FDA) [91]. Carboxymethyl-dextran (CMD) coated SPIONs were synthesized to connect with anti-CD44 antibody, whose contents of CMD influence its stability [92]. Dual-crosslinked amine activated dextran (AMD) coated SPIONs increased molecular weight of the coating and improved the dispersion [93]. Dextran coated INOPs were treated with epichlorohydrin to prevent dextran dissociation, which could produce cross-linked iron oxide nanoparticles (CLIO) with a high circulation half-life in blood and no acute toxicity [94].

Dextran and its modifications are used in the study of cerebral disease. Stem cell therapeutics for PD focus on the differentiation of mesenchymal stem cells (MSCs) to dopaminergic (DA) producing cells, replacing damaged DA neurons in PD. Dextran-coated or dextran-derivatives-coated iron oxide (IO) nanoparticles (NPs) (Dex-IO NPs) can activate the tropism of human MSCs (hMSCs) from bone marrow toward tumor [95]. A combination of dextran sulfate coated INOPs and quercetin was less toxic to PC12, a cell line readily quantifiable, rapid, and reversible response to nerve growth factor (NGF) [96], than dimercaptosuccinic acid (DMSA)-coated INOPs. In detail, dextran sulfate coated INOPs with concentration less than 50 μg/mL had no significant toxicity to PC12 cells while 1.5 mM DMSA modified INOPs were very toxic to PC12 cells [97].

#### 2.2.2. Chitosan

Chitosan (CS) is another important polysaccharides coating, providing positive charge for cell adsorption and opening tight junctions between epithelial cells, thus facilitating transportation [98]. Sterically stabilized chitosan-coated iron nanospheres were obtained after synthesis because of the low solubility of chitosan in the pH of INOPs synthesis [99,100]. Chitosan-based INOPs enhanced gene transfection as gene delivery vector [101]. O-carboxymethyl chitosan (CC) bound Fe_3_O_4_ was developed, improving the problem of chitosan’s bad solubility [102]. Trimethyl chitosan (TMC) coated SPIONs, TMC and CMD coated INOPs were also produced to delivery siRNA. They found that CMD-TMC-INOPs had little improvement effect on cytotoxicity and siRNA load compared with TMC-INOPs [103]. This may suggest that a monolayer surface coating of polysaccharide is enough for drug circulation. However, dextran-INOPs were further coated by chitosan to increase surface charge [104], indicating the importance of the order in the multilayer coating.

A combinatorial nano-system was made to treat glioblastoma multiforme (GBM). With tumor-specific ligand-transferrin (Tf) as target, CS-INOPs containing antitumor drug doxorubicin (DOX) for cancer cell killing and fluorescent dye Rhodamine B isothiocyanate (RBITC) for simultaneously intracellular fluorescent showed improved cell uptake and cell killing through a concurrence of cell apoptosis and autophagy in the treated tumor U251 cells. The saturation magnetization of INOPs was estimated to be 56.06 emu/g while CS/INOPs and Tf-CS/INOPs were 35.34 and 20.94 emu/g respectively, demonstrating that the saturation magnetization decreased after coating. The apoptosis of U251 MG cells treated with Tf-CS/INOPs were about 70% and the cell viability came down to about 55% [105]. Another bimodal molecular imaging system, overcoming the cons of magnetic (low sensitivity) and fluorescence (low resolution) imaging was made by combining CS-INOPs with CdS. Podophyllotoxin (PD), a naturally occurring antimitotic agent, was added, too. Deposition of chitosan NPs in lungs no other organs may ascribe to the mucoadhesion property of chitosan [106], which may be resolved by CS modification with other biocompatible molecule.

#### 2.2.3. Poly (Ethylene Glycol) PEG

PEG, as a amphiphilic molecule at the size lower than 100 kD [107], is synthesized by anionic ring opening polymerization of ethylene oxide, regarded to be “stealth” molecule to reduce reticuloendothelial (RES) clearance [108], reduce nanoparticle uptake by macrophages [109,110] and extend blood circulation time in vivo, making it suitable for targeted therapy. PEG deposits on the surface of INOPs in situ [111] or grafted PEG connects to INOPs via middle coating, like chitosan, silane [112]. The development of dual function PEG expands its application in coating.

A MGMT inhibitors O6-benzyl guanosine (BGS) covalently attached to PEG-INOPs in a pH-response manner to improve cell sensitivity to TMZ. The combination reduced the cell resistance by approximately 40-fold while BGS reduced resistance by 19-fold, demonstrating BGS-PEG-INOPs’ greater sensitivity to TMZ in human GBM cells [113]. Methotrexate (MTX), a short half-life and rapid diffusion drug for various forms of cancer, was covalently bound to Fe_3_O_4_ nanoparticles via a bifunctional PEG with the end of silane and trifluoroethylester (TFEE). Intracellular uptake of INOPs-PEG-MTX was higher than NP-dextran in a concentration-dependent manner in glioma [114]. Moreover, PEG-PEI copolymer coated INOPs loaded Dox in a pH-response manner to test the ability of reducing drug efflux in glioma. The IC_50_ of INOPs-Dox in C_6_-ADR glioma was approximately 3- to 5-times lower than that of free DOX in cells, demonstrating that preventing drug efflux from drug-resistant cells caused enhanced cell death [115]. PEI-PEG-IONPs loaded salinomycin, an anticancer drug, decreased the glioblastoma cell viability to 45% compared with free-salinomycin’s 36% [116].

#### 2.2.4. Polyethyleneimine (PEI)

PEI with highly positive charge is used for gene delivery [117,118]. It has primary, secondary, and tertiary amines, of which two-thirds of the amines can be protonated in a physiological environment, helping escape from endosome due to proton sponge effect and release drugs to cytoplasm [119]. PEI has been widely used in DNA or siRNA gene transfection in vitro [120]. PEI-coated INOPs can be made by in situ, post-synthesis adsorption, or post-synthesis grafting [107]. However, PEI exerts cytotoxic effects on living organisms and short half-life in blood due to non-specific interactions with negatively charged cell-surface, which needs to be considered in its application.

PEI-INOP was made to determine its applicability in gene therapy of brain cancer. The result exhibited no difference in cell toxicity compared with slightly anionic G100 but lower blood concentration [121]. After folic acid-conjugated, cellular uptake was increased [122]. Another research used citraconic anhydride to block primary amine group of PEI, eliciting a pH-sensitive cytotoxic effect in the acidic tumor microenvironment. The study demonstrated lower toxicity of PEI to cell in pH 7.4. Gene silencing effect of siRNA in pH 7.4 and pH 6.2 was 93.8% and 50.8% respectively, making it a pH-response gene silencing nanoparticles (NPs) [123]. INOPs, coated with PEG-grafted chitosan and PEI, functionalized with chlorotoxin (CTX), were used to delivery siRNA. Cells, treated with NP/PEG-CS/PEI-siRNA, exhibited a 53.5% reduction in gene expression compared with a reduction of 88.7% in no siRNA, making NP/PEG-CS/PEI-siRNA a platform for glioma gene therapy [124]. A high drug-loading system was made by combining PEI with phenylboronic acid (PBA) to treat glioma. The Dox loading of PBA-PEI-INOPs was 2.26 and 3.27 times greater than that of PBA-INOPs and PEI-INOPs [125]. Based on PEI, the cationic redox-sensitive PEI (SSPEI) was developed to reduce cytotoxicity. The cell viability of SSPEI-INOPs-pDNA was 8 times than that of PEI and gene transfer efficiency was a little better than PEI in the presence of magnet, showing better biocompatibility in no reduction of transfection efficiency [126].

#### 2.2.5. Poly (Carboxybetaine) PCB

PCB has both cationic and anionic groups, endowing it with the charge-reversible ability in different pH. It is neutral in physiological condition, resisting nonspecific protein adsorption, which changes to positive charge via protonation in acidic environment, helping escape from lysosome [127]. These characteristics make it a special coating for INOPs.

Simvastatin covalently connected ROS-response PCB (Sim-se-se-PCB) was constructed to load hydrophobic SPIONs and absorb negative let-7b antisense oligonucleotide, reducing the differentiation of exogenous neural stem cells (NSCs) and enhancing the secretion of brain-derived neurotrophic factor (BDNF) synergistically. The transfection efficiency of PCB-Se-Se-Sim/let-7b NPs was comparable to that of Lipofectamine 2000 and significantly (170.8%) increased the BDNF level in the culture medium, which further induced the proliferation of NSCs and rescued the memory deficits in 2× Tg-AD mice [128].

Based on PCB, other copolymers were designed to form INOPs-loaded micelles. PCB-poly (2-hydroxyethyl methacrylate)-poly (carboxybetaine) (PHEMA-PCB) polymers were constructed to coat INOPs and delivery retinoic acid (RA) and siRNA (SOX-9), reducing the inhibition the effect of retinoic acid. NPs caused knock-down of approximately 52.3% of SOX-9 mRNA compared with negative controls and significantly improved the learning and memory of AD mice [129] (Figure 5).

#### 2.2.6. PLGA

PLGA is a copolymer of poly lactic acid (PLA) with poly glycolic acid (PGA). It is highly biocompatible and has been approved by FDA as drug delivery vehicles [130]. Various types of block copolymers of PLGA with PEG have been developed, such as PLGA-PEG, PEG-PLGA-PEG [130]. PEG-PLGA coated INOPs were used to delivery PTX as anti-glioma drug. NPs accumulated in liver, but aspartate aminotransferase (AST) level returned to normal after a week, indicting the biocompatibility of PEG/PLGA-INOPs. PTX-PEG/PLGA-INOPs showed less toxicity than PTX [131].

Polymers provide a promising future in the application of INOPs in cerebral disease after targeting and drugs loading. However, the coating materials may cause low magnetic properties. And the blood concentration of NPs is low, which may cause disassembly of micelle before reaching the target.

### 2.3. Lipid Molecule Coating

Lipid, as the constituent of cellular membranes, provides a biocompatible protective barrier for nanoparticles. It can form a closed double layer structure in aqueous solution containing hydrophilic and hydrophobic space. There are two embedding methods for iron nanoparticles, in the phospholipid bilayer or in the core (Figure 6). Research found that these magnetic liposomes had high resistance against intracellular degradation compared with dextran- or citrate-coated particles [132]. Adding PEG-modified phospholipid provides an opportunity for drug, target, or other functional groups binding [133,134] and prolongs blood circulation time, facilitates tumor accumulation via enhancing permeability and retention (EPR) effect [135]. Zwitterionic polymer poly(carboxybetaine) (PCB) modified distearyl phosphatidylethanolamine (DSPE) increased the cell uptake and promoted lysosomal drug release compared with PEG [136].

Numerous studies have been done on the brain application of DSPE-PEG based iron liposome. DSPE-PEG-B6, high affinity for transferrin receptor (TfR), DSPE-PEG-Mazindol (MA), a high affinity dopamine transporter, DSPE-PEG-phenylboronic acid, loading epigallocatechin gallate (EGCG), and INOPs were self-assembled to treat PD. With the help of MA, NPs were internalized into dopaminergic neurons after crossing BBB with help of B6. In the ROS environment of dopaminergic neurons, EGCG played the role of anti α-synclein(αS) aggregation and promoting neurotransmitter circulation. INOPs functioned as MR imaging. NPs reduced αS aggregation compared to cells treated with only EGCG and obtained good therapeutic efficacy in PD [137]. Testing reagents and drugs have also been delivered by DSPE-PEG formed INOPs. Congo red, specifically detecting amyloid plaques that have extensive β-sheet structures, and rutin, a glycone of quercetin with the ability of interfering with Aβ aggregation and reducing neurotoxicity, were delivered by DSPE-PEG formed iron-contained liposomes, coinjected with mannitol to treat AD. The cell viability of Aβ^+^Cu^2+^ cells improved to 91% from 63% and had a greater contrast-to-noise ratio of MRI. NPs could significantly rescue memory deficits in AD transgenic mice [138]. Different from INOPs in the core, INOPs were produced in the hydrophobic region of phospholipid bilayer with quantum dots (QDs), a new optical imaging method for molecular tracing and biomedical diagnostics, and cilengitide (CGT), inhibiting overexpress of integrin receptors. Using ultrasound-targeted microbubble destruction for delivery, the median survival of glioma rat was significantly extended to 59.6d compared with the control’s (20d) [139].

Lipid coating provides more biocompatible modification for INOPs. Besides, it can encapsulate lots of INOPs and deliver them to the target site, avoiding dilution like micelle, resulting in a high concentration of drugs for therapy and INOPs for imaging.

### 2.4. Lipid and Polymers Coating

INOPs in the core of liposomes need to be hydrophilic. Polymers are one of the methods to change INOPs from hydrophilicity to hydrophobicity, which provide a chance for the combination of lipid and polymers. Moreover, different from supported lipid bilayers (SLB), the inner and the outer leaflet being composed of the same molecules, hybrid lipid bilayer (HLB) is composed of different molecules where the composition of the two leaflets are not identical and only the outer leaflet actually contains lipid molecules, adapting a lipid-bilayer-like arrangement [140]. The combination of lipid and polymers are also applied in HLB. In addition, amphiphilic lipid can also self-assembly into micelles with polymers. DSPE-PEG micelles have greater stability because of interaction of hydrophobic acyl chains in phospholipid fragments, making the CMC of DSPE-PEG micelle 100-fold lower than the CMC of polymeric micelles fabricated from other block copolymers [141,142]. More hydrophobic drugs are possible to be loaded because of the hydrophobicity conferred by the diacyl lipid chains [143]. The addition of lipid in polymers increased the biocompatibility of NPs.

Three drugs, Dox, combretastatin A4 (CA4), and all-trans retinoic acid (ATRA) were combined through pH-sensitive poly (β-amino ester) (PAE), hypoxic-response azobenzene (AZO) and hydrophobic interaction together with IONPs delivered in the core of DSPE-PEG and PAE constituted micelle to release in order. CA4 was released in GBM acid environment to destroy the angiogenesis after B6 mediated BBB crossing. Then ATRA and Dox played their roles of enhancing GSCs differentiating to glioblastomas and killing them after PAE-mediated lysosomal escape. NPs extended the circulation time of the drugs and maintained survival rate at 60% after treating compared with dying in other groups [144]. After the development of PCB with good lysosomal escape ability, DSPE-PCB based lipid envelope was constructed. Focusing on the immunosuppressive tumor microenvironment at the same time of chemotherapy in glioblastoma, INOPs were encapsulated in the core with siRNA (TGF-β) on the hydrophilic surface of the micelle formed by an ROS-responsive polymer poly[(2-acryloyl) ethyl (p-boronic acid benzyl) diethylammonium bromide] (BA-PDEAEA, BAP) before coated with TMZ loaded DSPE-PCB liposome. After the ANG-2 mediated BBB crossing and PCB-mediated lysosomal escaping, TMZ and TGF-β were released to kill tumors. The NPs reduced TGF-β by 44% and maintained 80% survival while the others died [145] (Figure 7).

### 2.5. Other Coatings

Other materials are also used to improve iron nanoparticles performance [146]. Au coating was reported to enhance imaging signals and endow the ability of target and function modification [147]. Metal oxides can protect the magnetic nuclei from oxidation, for example, the MgO shell can protect magnetic nuclei from oxidation up to 600 °C [140].

The negatively charged polydopamine (PDA) coated Fe_3_O_4_ was coated with positively charged p53 plasmid-loaded poly (2-dimethyl amino) ethyl methacrylate (PDM)-coated Au through consecutive electrostatic assembly. The C6 glioma cell viability decreased to 24.1% by combining PTT with gene therapy in vitro and produced an effective inhibition of high-proliferation-rate C6 glioma in vivo [148] (Figure 8).

## 3. Conclusions and Prospection

Iron-based nanoparticles are promising targeted drug carriers for theranostics of brain diseases. After surface modification, they have the abilities to (1) encapsulate multiple therapy molecules, like gene, protein, chemical drugs to achieve multi-drug cooperation therapy; (2) deliver drugs to target sites to increase drug concentration in disease site; (3) release loaded drugs according to various stimuli conditions of disease site. There have been FDA-approved INOPs [149,150]. For example, Feraheme^®^ has been approved for iron deficiency anemia in adults [4], which show advantages over other nanoparticles as a component of human body. Ferucarbotran, Ferumoxide, and Ferumoxytol have been approved for central nervous system (CNS) imaging [15], showing the application potential of delivery in central nervous system diseases.

INOPs approved by FDA are in the first generation-imaging. Later, NanoTherm was approved to treat recurrent glioblastoma multiforme in 2010 [15]. There are no FDA-approved INOPs for drug delivery yet. The first consideration is the targeting. Target ligands may not always work because of their deposition in liver and kidney organs when administered intravenously. To solve the problem, further development of administration mode and surface coating will be needed, which can reduce the macrophage-induced phagocytosis and protein interaction. It is exciting that studies have found that therapeutic agents can be delivered to the brain by convention enhanced delivery (CED) at high concentration without much toxicity to normal organs [151]. The second consideration is the drug leakage. Common polymers micelle may be in a state of disassembly when the blood concentration of polymer is below CMC after drug injection, causing quick drug release. In this regard, crosslinked coatings offer a solution [152,153]. Moreover, the development of iron-based nanoparticles largely depends on uniformity and size of SPIONs.

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
