# Peer review of "Surface Modification of Iron Oxide-Based Magnetic Nanoparticles for Cerebral Theranostics: Application and Prospection"

_nanomaterials, 2020, doi:10.3390/nano10081441_

Round 1

Reviewer 1 Report

The manuscript suffers from very poor English. The sentences are not built correctly, but an even bigger problem is that the concepts are introduced without appropriate definitions. 

I would like to remind to the authors that the purpose of a review is provide the reader with a concise information on a subject without the necessity to look for explanations elsewhere.  

Author Response

Response to Reviewer 1 Comments

Point 1: The manuscript suffers from very poor English. The sentences are not built correctly, but an even bigger problem is that the concepts are introduced without appropriate definitions.   

Response 1: Thank you for comments. We have made correction and marked with red color in manuscript.

Examples of the revisions in the manuscript:

(1) Page 2, line 47, “Blood-Brain Barrier (BBB), as a protective barrier for central nervous system, limits uptake of drugs for brain disease.” change to “As a protective barrier for central nervous system, Blood-Brain Barrier (BBB) limits uptake of drugs for in brain disease.”

(2) Page 4, line 113, “Nanomedicine opens new gate for glioma, overcoming BBB.” change to “Nanomedicine opens a new gate for glioma in overcoming BBB.”

(3) Page 5, line 138, “They also perform well as tail of copolymers like PEG, PEI.” change to “They also perform well as tail of copolymers like Poly (ethylene glycol) (PEG), Polyethyleneimine (PEI).”

(4) Page 8, line 298, “reducing the differentiation of exogenous NSCs and enhancing the secretion of BDNF synergistically.” change to “reducing the differentiation of exogenous neural stem cells (NSCs) and enhancing the secretion of (brain-derived neurotrophic factor (BDNF) synergistically.”

Point 2: I would like to remind to the authors that the purpose of a review is provide the reader with a concise information on a subject without the necessity to look for explanations elsewhere. 

Response 2: Thank you for comments. We will pay more attention to it in our later work. At the same time, we try to summarize the general rule from researches, which we think is crucial for avoiding repetitive work.

Reviewer 2 Report

The Review is comprehensive, fair job made by the authors, and it takes in account a substantial amount of data in the area. Interesting reading, especially for a person not deeply immersed into this very challenging area. But if a Review is intended to educate a wide audience, including non-specialists, some specific shortcomings which must be overcome:

1. The Abstract as it stands doesn’t really reflect the actual contents of the manuscript (which is primarily focussed on various nanoparticle coatings, but not clinical applications as claimed, e.g. “we focus on the application of various coated INOPs in brain disease”). It perhaps comes from the fact that the manuscript must have been written mostly by the highly qualified Chemists with unfortunately, a limited specialization into medicine. The English of the Abstract must be also improved - this is your showcase, make it perfect!

2. The Review must be more focussed on critical opportunities into the key unresolved issues of MNPs THERANOSTICS, as claimed. One of these is the fact that besides the MNP diagnostic modalities or MNP-enabled treatments on their own, NO clinically approved nanotools still exist to treat the patients! This is a key issue, and it MUST be much strongly addressed in the context of what can be done to overcome this hurdle and to move on to the next stage.

3. Please deliberate more on the spreading of injected NPs vs their systemic intravenous injections, since this is a very hot issue, and nobody can claim today that the therapeutically intended doses on applied/injected MNP do not impose excessive harm for the human body overall and individual organs.

4. Conclusions and prospection are too brief, must be expanded and need more discussion on how to overcome the existing clinical translation gap with actual drug delivery issues and the well-known now a problem of true multi-functionality of nanomaterials.

5. The Authors must be the enthusiasts of the depicted idea in general, and a lot has been done in the labs over the past years in the area, but where in the timeline for a single multi-functional theranostic magnetic MNP approved? – This must be the bottom line of the critical summary of this review. Give us a hope, or otherwise why do we work for it?

Author Response

Response to Reviewer 2 Comments

Point 1: The Abstract as it stands doesn’t really reflect the actual contents of the manuscript (which is primarily focussed on various nanoparticle coatings, but not clinical applications as claimed, e.g. “we focus on the application of various coated INOPs in brain disease”). It perhaps comes from the fact that the manuscript must have been written mostly by the highly qualified Chemists with unfortunately, a limited specialization into medicine. The English of the Abstract must be also improved - this is your showcase, make it perfect!   

Response 1: Thanks for your comments. According to your advice, we have changed the abstract in the manuscript.

The revisions in the manuscript:

Combining diagnosis with therapy, magnetic iron oxide nanoparticles (INOPs) act as an important vehicle for drug delivery. However, poor biocompatibility of INOPs limits their application. To improve the shortcomings, various surface modifications have been developed. Small molecules coatings retain the original magnetic properties and improve the hydrophilicity at the same time. Polymers coatings endow nanoparticles the ability to target and release drugs in control. Lipid coatings prolong blood circulation time via reducing cell interaction. These surface coatings allow nanoparticles to be modified to cross BBB, which is essential for diagnosis and treatments of brain disease. Here we focus on the characteristics of different coated INOPs and their application in brain disease, particularly gliomas, Alzheimer's disease (AD) and Parkinson's disease (PD). Moreover, we summarize the current progress and expect to provide help for future researches.

Point 2: The Review must be more focused on critical opportunities into the key unresolved issues of MNPs THERANOSTICS, as claimed. One of these is the fact that besides the MNP diagnostic modalities or MNP-enabled treatments on their own, NO clinically approved nanotools still exist to treat the patients! This is a key issue, and it MUST be much strongly addressed in the context of what can be done to overcome this hurdle and to move on to the next stage.

Response 2: Thanks for your comments. We have recently found that NanoTherm was approved to treat recurrent glioblastoma multiforme. However, there are no FDA-approved INOPs for drug delivery. And we modified the section of “Conclusion and prospection”.

The revisions in the manuscript:

Iron based nanoparticles are promising targeted drug carriers for BBB crossing and theranostics of brain diseases. After surface modification, they have the abilities to 1) encapsulate therapy molecule, like gene, protein, chemical drugs to achieve multi-drug cooperative therapy; 2) delivery drugs to targeted sites to increase drug concentration; 3) release loaded drugs according to various stimuli conditions of diseases’ sites. For example, Feraheme® have been approved to be used for iron deficiency anemia in adults [4], which show advantages over other nanoparticles as component of human body. Ferucarbotran, Ferumoxide, and Ferumoxytol have been approved for central nervous system (CNS) imaging [142], showing the possibility of delivery application in central nervous system diseases.

INOPs approved by FDA are in the first generation-imaging. NanoTherm was also approved to treat recurrent glioblastoma multiforme in 2010 [142]. There are no FDA-approved INOPs for drug delivery yet. The first consideration is the targeting. Targeting ligands are not always work because of their deposition in liver and kidney organs when administered intravenously. To solve the problem, further development of the mode of administration and surface coating will be needed, which can reduce phagocytosis and protein interaction. What's exciting is that studies have found therapeutic agents can be delivered to brain by convention enhanced delivery (CED) at high concentration without much toxicity to normal organs [143]. The second consideration is the drug leakage. Common polymers micelle may be in the state of depolymerization when the blood concentration of polymer is below CMC after drug injection, causing quickly drug released. In this regard, crosslinked coatings offer direction [144; 145]. What’s more, the development of iron-based nanoparticles largely depends on uniform size and size of SPIONs.

Point 3: Please deliberate more on the spreading of injected NPs vs their systemic intravenous injections, since this is a very hot issue, and nobody can claim today that the therapeutically intended doses on applied/injected MNP do not impose excessive harm for the human body overall and individual organs.

Response 3: Thanks for your comments. We have read relevant materials and found that targeting may not work because of deposition in liver and kidney organs intravenously. To solve this problem, new injection has been developed, like convention enhanced delivery (CED), as mentioned in the “conclusion and prospection” section.

Point 4: Conclusions and prospection are too brief, must be expanded and need more discussion on how to overcome the existing clinical translation gap with actual drug delivery issues and the well-known now a problem of true multi-functionality of nanomaterials.

Response 4: Thanks for your comments. We have modified the part of “conclusion and prospection” according to suggestions.

Point 5: The Authors must be the enthusiasts of the depicted idea in general, and a lot has been done in the labs over the past years in the area, but where in the timeline for a single multi-functional theranostic magnetic MNP approved? – This must be the bottom line of the critical summary of this review. Give us a hope, or otherwise why do we work for it?

Response 5: Thanks for your comments. We think that everything has a process especially drugs. There have been approved drugs for magnetic thermotherapy as mentioned in the article. We should develop it with more patience with the progress on surface coating and iron preparation.

Reviewer 3 Report

Authors have intensively described iron-based nanoparticles from size, coating and treatment on brain disease. However, the information of treatment  on brain disease, gliomas, Alzheimer's disease (AD), Parkinson's disease (PD) is very large. please include the size and coating of iron-based nanoparticles and use tables to summarise them.

Minor correction

functionalized with chlorotoxin (CTX) should be mentioned at line 140.

Line 144

Small molecules coating, such as catechol,  reduces loss of original magnetic properties to a great extent and can exerts antioxidant activity and improve solubility. [reference]

Author Response

Response to Reviewer 3 Comments

Point 1: Authors have intensively described iron-based nanoparticles from size, coating, and treatment on brain disease. However, the information of treatment on brain disease, gliomas, Alzheimer's disease (AD), Parkinson's disease (PD) is very large. please include the size and coating of iron-based nanoparticles and use tables to summarise them.  

Response 1: Thanks for your comments. It’s necessary to use a table to summarize. But according to the key of our article, we use a table including Coating Classification, Coating (examples), and Characteristics.

The revisions in the manuscript:

Table 1. Summary of different coatings.

Coating Classification

Coating (examples)

Characteristics

Small molecule

Catechol, carboxylic, phosphate, sulfate, citrate, silane

reducing loss of original magnetic properties, antioxidant activity

Polymer

Dextran, Chitosan, PEG, PEI, PCB, PLGA

magnetic reduction, providing positive charge, reduce reticuloendothelial (RES) clearance, proton sponge effect

Lipid molecule

DSPE-PEG, DSPE-PCB

biocompatible protective barrier

Polymer and lipid molecule

BAP+DSPE-PCB

PAE+DSPE-PEG

combination of lipid and polymers

Other coatings

Au, MgO

function modification

Round 2

Reviewer 1 Report

The attempt of the authors to improve the manuscript is appreciated but is unfortunately not sufficient. Even the changes introduced after revision contain multiple arrows. Jere are just very few examples: 

Page2: blood-brain barrier (BBB) limits uptake 55 of drugs in brain disease ("uptake in a disease"... uptake can be in cells, tissues etc but not in a disease)

Page4: Nanomedicine opens a new gate for glioma in overcoming BBB (is glioma overcoming BBB?)

Page 5: situ coating (in situ)

Page14: Targeting ligands are not always work (wrong grammar) 

Author Response

Response 1: Thank you for comments. We have made correction and marked with “Track Changes” in the manuscript and marked in red in the attachment.

Reviewer 3 Report

Authors have included a table to summary different coating. please include the references according the different coating presented in the table 1.

Author Response

 Response 1: Thanks for your comments. We have made corrections and showed it below.

The revisions in the manuscript:

Table 1. Summary of different coatings.

Coating Classification

Coating (examples)

Characteristics

Small molecule

Catechol, carboxylic, phosphate, sulfate, citrate, silane

reducing loss of original magnetic properties [72], antioxidant activity [78]

Polymer

Dextran, Chitosan, PEG, PEI, PCB, PLGA

magnetic reduction [142], providing positive charge [110], reduce reticuloendothelial (RES) clearance [101], proton sponge effect [120]

Lipid molecule

DSPE-PEG, DSPE-PCB

biocompatible protective barrier [125]

Polymer and lipid molecule

BAP+DSPE-PCB

PAE+DSPE-PEG

combination of lipid and polymers [134-136]

Other coatings

Au, MgO

function modification[133; 140]
